# TET2 Inhibits *PD-L1* Gene Expression in Breast Cancer Cells through Histone Deacetylation

**DOI:** 10.3390/cancers13092207

**Published:** 2021-05-04

**Authors:** Yinghui Shen, Lu Liu, Mengyuan Wang, Bo Xu, Ruitu Lyu, Yujiang Geno Shi, Li Tan

**Affiliations:** 1Center for Medical Research and Innovation, Shanghai Pudong Hospital, Fudan University Pudong Medical Center, and Shanghai Key Laboratory of Medical Epigenetics, Institutes of Biomedical Sciences, Fudan University, Shanghai 200032, China; 16111510017@fudan.edu.cn (Y.S.); 18111510045@fudan.edu.cn (L.L.); 17211510019@fudan.edu.cn (M.W.); 20211510009@fudan.edu.cn (B.X.); 2Department of Chemistry, University of Chicago, Chicago, IL 60637, USA; lvruitu@uchicago.edu; 3Division of Endocrinology, Diabetes and Hypertension, Brigham and Women’s Hospital, Harvard Medical School, Boston, MA 02115, USA

**Keywords:** TET2, PD-L1, HDAC, breast cancer, epigenetic regulation

## Abstract

**Simple Summary:**

Programmed cell death ligand 1 (PD-L1) is an essential immune checkpoint molecule that helps tumor cells to escape the immune surveillance. The aim of the current study was to investigate the epigenetic mechanisms underlying the aberrant expression of PD-L1 in breast cancer cells. Here, we identified TET2 as a negative regulator of PD-L1 gene transcription in breast cancer cells. Mechanistically, TET2 recruits HDAC1/2 to the PD-L1 promoter and facilitates the deacetylation of H3K27ac, resulting to the suppression of PD-L1 gene transcription. Our work reveals an unanticipated role of TET2-HDAC1/2 complex in the regulation of PD-L1 gene expression, providing new insights into the epigenetic mechanisms that drive immune evasion during breast cancer pathogenesis.

**Abstract:**

Activation of PD-1/PD-L1 checkpoint is a critical step for the immune evasion of malignant tumors including breast cancer. However, the epigenetic mechanism underlying the aberrant expression of PD-L1 in breast cancer cells remains poorly understood. To investigate the role of TET2 in the regulation of PD-L1 gene expression, quantitative reverse transcription PCR (RT-qPCR), Western blotting, chromatin immunoprecipitation (ChIP) assay and MeDIP/hMeDIP-qPCR were performed on MCF7 and MDA-MB-231 human breast cancer cells. Here, we reported that TET2 depletion upregulated PD-L1 gene expression in MCF7 cells. Conversely, ectopic expression of TET2 inhibited PD-L1 gene expression in MDA-MB-231 cells. Mechanistically, TET2 protein recruits histone deacetylases (HDACs) to PD-L1 gene promoter and orchestrates a repressive chromatin structure to suppress PD-L1 gene transcription, which is likely independent of DNA demethylation. Consistently, treatment with HDAC inhibitors upregulated PD-L1 gene expression in wild-type (WT) but not TET2 KO MCF7 cells. Furthermore, analysis of the CCLE and TCGA data showed a negative correlation between TET2 and PD-L1 expression in breast cancer. Taken together, our results identify a new epigenetic regulatory mechanism of PD-L1 gene transcription, linking the catalytic activity-independent role of TET2 to the anti-tumor immunity in breast cancer.

## 1. Introduction

For the past few years, immunotherapy has emerged as a frontline treatment for multiple malignancies, and now joins the ranks of surgery, radiation, chemotherapy, and targeted therapy for cancer therapy [1]. Different from traditional therapies, cancer immunotherapy utilizes the body’s own immune system to fight against tumor cells [2]. Among various immunotherapeutic strategies, the immune checkpoint blockade (ICB) has the broadest impact and prospects, with several antibodies targeting CTLA4 (cytotoxic T lymphocyte antigen 4), PD1 (programmed cell death 1), and PD-L1 (PD-1 ligand 1) approved by the FDA for the treatment of a number of different cancers [3,4,5]. PD-L1, encoded by the *CD274* gene, is an essential immune checkpoint molecule that is mainly expressed on the surface of tumor cells and macrophages [6]. The expression of PD-L1 is commonly elevated in cancer cells [7,8]. Cancer cells may exhibit immune escape upon recognition of PD-L1 by PD-1, which mediates T cell exhaustion [9]. Therefore, understanding the regulatory mechanisms of *PD-L1* gene expression in cancer cells is of great importance for improving responsiveness to anti-PD-L1 immunotherapy and suppressing immune evasion.

Many studies have revealed the transcriptional regulatory mechanisms of the *PD-L1* gene, including the inflammatory cytokines, specific transcription factors, and so on [10,11,12,13]. In addition, the epigenetic modifiers also play an important role in regulating *PD-L1* gene transcription, altering the chromatin accessibility for the transcription factors through DNA methylation or histone modifications [14,15,16,17]. For instance, DNA methylation at the promoter region is commonly considered to be an epigenetic mechanism of transcriptional silencing [18]. In contrast, TET (Ten-eleven translocation) family members, including TET1, TET2, and TET3, have been regarded as DNA demethylases for gene activation [19,20]. TET proteins initiate active or passive DNA demethylation by promoting 5mC (5-methylcytosine) oxidization [21]. Recent studies have shown that TET2 could act as a critical player in the regulation of immune homeostasis and anti-tumor immunity [22,23,24,25,26,27]. Although a recent work reported that TET2 augments the IFN-gamma-induced PD-L1 expression in melanoma, colon cancer, and acute monocytic leukemia cells [15], whether TET2 is involved in the epigenetic regulation of *PD-L1* gene expression in breast cancer remains largely unknown.

In this study, we set out to investigate the relationship between TET2 and PD-L1 in breast cancer. Surprisingly, we found that TET2 is a suppressor of *PD-L1* gene transcription in breast cancer cells. Mechanistically, TET2 inhibits *PD-L1* gene expression through recruiting HDAC1/2 to *PD-L1* gene promoter and facilitating the histone deacetylation, which is likely independent of DNA methylation and hydroxymethylation.

## 2. Materials and Methods

### 2.1. Cell Cultures

MCF7 and MDA-MB-231 human breast cancer cells were cultured in a DMEM high glucose medium (Hyclone, Marlborough, MA, USA) supplemented with 10% fetal bovine serum (FBS)(BI) and 1% penicillin/streptomycin (100 U/mL, Hyclone). All cells were cultured at 37 °C in a humidified incubator containing 5% CO_2_. The identity of two cell lines was confirmed by STR genotyping analysis.

### 2.2. Stable Cell Lines Construction

*TET2* knockout MCF7 cells were generated by the CRISPR method as described previously [28]. Mono-cell colonies were picked and expanded for identification and future experiments. The TET2 knockout efficiency of these colonies was examined by Western blot analysis of TET2 protein expression and PCR analysis of genomic DNA for indels around the sgRNA targeting region. The following target sequences were used for sgRNA design:

TET2KO sg#1: AGGACTCACACGACTATTC

TET2KO sg#2: GGAGAAAGACGTAACTTCG

Mock (empty vector) and *TET2*-overexpressing (O/E) MDA-MB-231 cells were generated using the *piggybac* system as described previously [29]. In brief, MDA-MB-231 cells were co-transfected with pCMV-PBase and *piggybac* plasmids (pPB-CAG-ires-Pac empty vector or pPB-CAG-*Flag-HA-TET2*-ires-Pac). Puromycin (2 μg/mL) was added to the culture medium at 48 h post-transfection. The puromycin-resistant stable cell colonies were picked and expanded for identification and future experiments.

### 2.3. Reagents and Antibodies

The following cytokines and chemical inhibitors were used for the treatment of cells in indicated experiments: IFN-gamma (Peprotech, Cranbury, NJ, USA, #300-02), 5-Aza-CdR (5-aza-2′-deoxycytidine, Sigma-Aldrich, St. Louis, MO, USA, #A3656), TSA (Trichostatin A, ApexBio, Houston, TX, USA, #A8183), and SAHA (Vorinostat, Selleck, Houston, TX, USA, #S1047).

The following antibodies were used for Western blot, ChIP, MeDIP and hMeDIP in our study: LaminB1 (Proteintech, #66095-1-Ig), TET2 (CST, Boston, MA, USA, #18950S), PD-L1 (CST, #13684S), GAPDH (Abclonal, Wuhan, HB, China, #AC033RRID), H3K27ac (Active Motif, Carlsbad, CA, USA, #39133), H3K4me3 (CST, #9751), H3K27me3 (CST, #9733), HDAC1 (CST, #34589S), HDAC2 (CST, #57156S), 5hmC (Active Motif, #39999), and 5mC (Active Motif, #39649).

### 2.4. Knocking Down by shRNAs

The oligos for TET2 shRNAs were subcloned into the pLKO.1-TRC vector (Addgene, Watertown, MA, USA). The shRNA lentiviral particles were packaged in 293T cells according to the manufacturer’s guidelines. MCF7 cells were infected with each lentivirus supernatant in the presence of 8 μg/mL polybrene. At 48h post infection, the infected cells were cultured with 2 µg/mL puromycin. After continuous puromycin screening for 5 days, the survival MCF7 cells were collected for RNA isolation and RNA-seq.

### 2.5. RT-qPCR

RNA was extracted from the cells using TRIzol reagent (Thermo Fisher Scientific, Waltham, MA, USA) according to the manufacturer’s protocol. The protocol of RT-qPCR was described in our previous study [13]. Gene expression levels were normalized to *GAPDH*. The sequences of primers were shown in Appendix A.

### 2.6. Western Blot

Cells were harvested and lysed with 1x SDS loading buffer. After quantification of the lysates with BCA assay, the Western blot assay was performed as described previously [13]. The primary antibodies used in this study were TET2 (1:1000), PD-L1 (1:1000), laminB1 (1:5000), H3K27ac (1:2000), and GAPDH (1:10,000).

### 2.7. ChIP-qPCR

The Chromatin immunoprecipitation (ChIP) assay was performed as previously described [29]. The enrichment levels of TET2, H3K4me3, H3K27me3, H3K27ac, and HDAC1/2 at *PD-L1* promoter were quantified by qPCR analysis of the ChIP products and the relevant inputs. Primer sequences used for ChIP-qPCR are listed in Appendix A.

### 2.8. MeDIP/hMeDIP-qPCR

Genome DNA was prepared from cells using DNeasy Blood & Tissue kit (QIAGEN). MeDIP/hMeDIP-qPCR was performed as described previously [30]. Enrichment of 5mC (5-methylcytosine) or 5hmC(5-hydroxymethylcytosine) at *PD-L1* promoter was quantified by qPCR analysis. Primer sequences used for MeDIP/hMeDIP-qPCR are listed in Appendix A.

### 2.9. Co-Immunoprecipitation

The co-immunoprecipitation assay was conducted as previously described [31].

### 2.10. RNA seq Analysis

RNA samples from MCF7 cells (WT, TET2 KO #1, TET2 KO#2, scramble, shTET2#1, and shTET2#2) were subjected for RNA-seq using the Illumina platform. First, raw reads were trimmed to remove adapters and low-quality bases using the trim_galore program (version 0.6.6) with parameters: “--paired --fastqc”. Secondly, trimmed fastq files were aligned to the human reference genome (hg19.fa from UCSC) with the tophat program (v2.1.1) with default parameters. We used the FPKM_count.py (RSeQC) to calculate FPKM (fragments per kilo base of transcript per M) to represent the abundance of gene expression. The bedgraph files were uploaded on UCSC genome browser for visualization. RNA-seq data has been deposited in the NCBI Gene Expression Omnibus (GEO) under the accession number GSE164032 (https://www.ncbi.nlm.nih.gov/geo/query/acc.cgi?acc=GSE164032, Submitted on 30 December 2020, Released on 1 June 2021).

### 2.11. Analysis of CCLE and TCGA Data

The gene expression data of 23 kinds of cancer cell lines were downloaded from CCLE (Comprehensive Cell Line Encyclopedia, http://www.broadinstitute.org/ccle/home; accessed on 2 May 2018). The expression levels of *TET2* and *PD-L1* from breast invasive carcinoma (Agilent microarray data) were downloaded from TCGA (The Cancer Genome Atlas, http://www.cbioportal.org; accessed on 19 March 2020). The *TET2* and *PD-L1* mRNA levels of cancer cell lines and breast cancer tissues were used for correlation analysis and linear regression analysis.

### 2.12. Statistical Analysis

GraphPad Prism Software was used for quantitative data visualization and statistical analysis. All graphs were presented as an average of at least three independent experiments. Standard deviation (S.D.) was calculated and presented as error bars in graphs. Comparisons between two groups were analyzed by paired Student’s *t*-test. Multiple comparisons were analyzed by one-way ANOVA with Tukey post-test. Significance level was set as *p* = 0.05 and presented as * in all graphs (*, *p* < 0.05; **, *p* < 0.01; ***, *p* < 0.001; ****, *p* < 0.0001; ns, not significant).

## 3. Results

### 3.1. TET2 Is a Negative Regulator of PD-L1 Gene Transcription in Breast Cancer Cells

In an RNA-seq analysis for the downstream genes of TET2 in MCF7 cells, we identified that *PD-L1* mRNA expression level was upregulated in *TET2* KO MCF7 cells compared to wild type (WT) MCF7 cells (Figure 1A). RT-qPCR analysis also confirmed the substantial increase in *PD-L1* mRNA levels in *TET2* KO MCF7 cells (Figure 1B). Given that the basal expression level of *PD-L1* is relatively low in MCF7 cells, we also treated cells with IFN-gamma and measured the impact of TET2 depletion on *PD-L1* expression. RT-qPCR and Western blotting analyses showed that IFN-gamma treatment dramatically upregulated *PD-L1* expression in both WT and *TET2* KO MCF7 cells (Figure 1C,D). Importantly, regardless of the absence or presence of IFN-gamma stimulation, the mRNA and protein levels of *PD-L1* in *TET2* KO MCF7cells were correspondingly higher than those in WT MCF7 cells (Figure 1C,D). Furthermore, *PD-L1* intron mRNA level displayed a similar increase in *TET2* KO MCF7 cells (Appendix A), indicating a direct effect of TET2 depletion on nascent RNA synthesis rather than mRNA stability. Knocking down *TET2* by shRNA also increased the *PD-L1* mRNA level in MCF7 cells (Appendix A). Moreover, the MDA-MB-231 cell is a commonly used triple-negative breast cancer cell line in laboratory research. Compared to MCF7 cells, MDA-MB-231 cells express relatively low levels of TET2 expression but high levels of PD-L1 expression, showing features of an ideal model for a TET2 “gain-of-function” study. To verify the suppressive role of TET2 on PD-L1 expression, we overexpressed TET2 in MDA-MB-231 cells and observed the downregulation of *PD-L1* gene expression upon TET2 overexpression (Appendix A). Overall, our data demonstrate that TET2 functions as a negative regulator of *PD-L1* gene transcription in breast cancer cells.

### 3.2. TET2 KO Does Not Alter the DNA Methylation and Hydroxymethylation Level at the Promoter Region of PD-L1 Gene

Next, we investigate the molecular mechanism through which TET2 inhibits *PD-L1* gene transcription in breast cancer cells. By analyzing the published TET2 ChIP-seq data (GSE120756) [32], we discovered that TET2 directly bound to the promoter region of *PD-L1* gene in MCF7 cells (Figure 2A). Our TET2 ChIP-qPCR data also confirmed the occupancy of TET2 at the *PD-L1* promoter, indicating a direct action of TET2 on *PD-L1* gene transcription (Figure 2B). Since TET2 has the ability to catalyze 5mC oxidation and initiate DNA demethylation, we examined the 5mC and 5hmC enrichment at the promoter region of *PD-L1* gene by MeDIP- and hMeDIP-qPCR. Unexpectedly, TET2 depletion has no significant effect on the 5mC and 5hmC enrichment at *PD-L1* promoter region (Figure 2C,D). Given the critical role of TET2 in DNA demethylation, TET2 KO may increase DNA methylation levels on other genomic regulatory regions beyond the *PD-L1* promoter, which might contribute to the transcriptional activation of *PD-L1* gene. To exclude this possibility, we treated WT and TET2 KO MCF7 cells with 5-Aza-CdR (a DNMT inhibitor) and examined the mRNA expression of the *PD-L1* gene. Figure 2E showed that 5-Aza-CdR treatment was not sufficient to change PD-L1 expression in either WT or TET2 KO MCF-7 cells. Thus, our data suggest that TET2-mediated inhibition of *PD-L1* gene transcription is not dependent on its 5mC dioxygenase activity.

### 3.3. TET2 Recruits HDAC1/2 to Deacetylate H3K27ac at PD-L1 Promoter

In addition to the DNA-demethylating activity, TET2 can also function as a transcriptional co-factor to regulate gene expression [33]. We profiled the well-studied histone modifications (H3K4me3, H3K27me3, and H3K27ac) at the *PD-L1* promoter region in WT and *TET2* KO MCF7 cells. ChIP assay showed that the levels of H3K4me3 enrichment at *PD-L1* promoter were comparable between WT and *TET2* KO MCF7 cells (Figure 3A), while the repressive H3K27me3 was rare at the *PD-L1* promoter in WT and *TET2* KO MCF7 cells (Figure 3B). Consistent to the changes of *PD-L1* gene transcription, the H3K27ac levels at *PD-L1* promoter region were increased in MCF7 cells upon TET2 depletion (Figure 3C). Additionally, TET2 overexpression reduced H3K27ac enrichment at the *PD-L1* promoter region in MDA-MB-231 cells (Appendix A). Our data suggest that TET2 is able to modulate the H3K27ac level at the *PD-L1* promoter in breast cancer cells.

TET2 has been shown to recruit histone deacetylases (HDAC1/2) to specific gene loci, thereby mediating the transcriptional repression in immune cells [34,35]. Therefore, we performed ChIP-qPCR analysis of HDAC1 and HDAC2 in WT and *TET2* KO MCF7 cells. As expected, the binding of HDAC1 and HDAC2 to the *PD-L1* promoter in *TET2* KO MCF7 cells was significantly lower than that in WT cells (Figure 3D,E). Moreover, we conducted a co-IP assay for the interaction between TET2 and HDAC1/2 using anti-HDAC specific antibody. As expected, we detected a TET2 band in the anti-HDAC1 immunoprecipitants from MCF7 cells and TET2-O/E MDA-MB-231 cells by Western blotting (Figure 3F & Appendix A). Next, we chose two HDAC inhibitors (TSA and SAHA) to treat WT and *TET2* KO MCF7 cells. As shown in Figure 3G, both inhibitors increased the global H3K27ac level in WT and *TET2* KO MCF7 cells. Importantly, the *PD-L1* mRNA expression level was elevated by treatment with HDAC inhibitors only in WT cells but not in *TET2* KO MCF7 cells (Figure 3H). We also observed that HDAC1 binding at the *PD-L1* gene promoter was increased in TET2 O/E MDA-MB-231 cells (Appendix A). Taken together, our data suggest a working model in which TET2 unites HDAC1/2 to deacetylate H3K27ac and establishes a repressive chromatin structure at the *PD-L1* promoter (Figure 3I).

### 3.4. Negative Correlation between TET2 and PD-L1 Expression Levels in Breast Cancer

To investigate the clinical significance of the TET2/PD-L1 axis, we analyzed the relationship between *TET2* and *PD-L1* expression levels in online breast cancer data. CCLE data analysis showed that *TET2* mRNA expression levels are negatively associated with *PD-L1* mRNA expression levels in 57 breast cancer cell lines (*p* = 0.0031) (Figure 4A). Interestingly, by analyzing other types of cancer cell lines from CCLE, we found that at least two other kinds of cancer types (lung and soft tissue) had a significant negative correlation between *TET2* and *PD-L1* mRNA expression levels (Appendix A). We also analyzed the correlation between TET2 and PD-L1 by separating the CCLE breast cancer cell lines into three tumor subtypes and found that TET2 expression is negatively correlated with PD-L1 expression in luminal subtype (*p* = 0.0282) (Appendix A). In a similar manner, TCGA data analysis also showed that *TET2* expression levels had a negative correlation with *PD-L1* expression levels in breast cancer (*n* = 1904, *p* = 0.0065) (Figure 4B). Through an analysis of the four subtypes of TCGA breast cancer data, we found that *TET2* and *PD-L1* showed a negative correlation only in luminal B and Her2-enriched subtypes (Appendix A).

## 4. Discussion

In this study, we demonstrate that TET2 inhibits *PD-L1* gene expression in breast cancer cells either under baseline conditions or upon IFN-gamma stimulation. Conversely, a recent work from Xu et al. [15] showed that IFN-gamma-induced *PD-L1* gene expression was impaired by TET2 depletion in murine melanoma (B16-OVA), colon tumor cells (MC38), and human monocytic cells (THP-1). Mechanistically, they revealed that TET2 could be recruited by STAT1 to hydroxymethylate the *PD-L1* gene promoter and enhance its transcription in murine melanoma and colon tumor upon IFN-gamma stimulation. However, our data showed that the *PD-L1* promoter of breast cancer cells is in a DNA hypo-methylated status and that TET2 depletion does not alter the DNA methylation or hydroxymethylation levels at the promoter region of *PD-L1* gene. Moreover, we found that treatment with 5-Aza-CdR, a DNMT inhibitor, could not enhance *PD-L1* gene expression in either WT or *TET2* KO MCF7 cells, suggesting that TET2 may repress *PD-L1* gene expression in a catalytic-activity independent manner. These opposite results indicate that TET2-mediated regulation of *PD-L1* gene expression may be largely dependent on the cancer/tissue types.

In addition to the well-known DNA demethylation activity, TET proteins have been shown to recruit histone deacetylases (HDACs) and mediate transcriptional repression in immune cells [34,35,36]. Coincidentally, our study demonstrates that TET2 recruits HDAC1/2 to deacetylate H3K27ac at *PD-L1* promoter and results in the transcriptional suppression of *PD-L1* gene. Although our work identifies that TET2 acts as a scaffold protein for the negative regulation of *PD-L1* gene transcription in breast cancer cells, it is still unclear how TET2 itself is recruited to the *PD-L1* promoter. Since dozens of transcription factors have been identified to recruit TET2 to specific gene loci for epigenetic regulation in different types of tissues [37,38,39], we speculate that a specific transcription factor may be responsible for this task in breast cancer.

Previous reports have showed that the downregulation of TET2 expression and 5hmC levels is an epigenetic hallmark of multiple types of cancers [21,40,41,42]. Dysregulation of the TET2/5hmC pathway promotes epithelial–mesenchymal transition (EMT), chemotherapy resistance, proliferation, invasion, and metastasis during breast cancer pathogenesis [43,44,45]. Based on the afore-mentioned regulatory mechanism, we speculate that TET2 loss may facilitate immune evasion by breast cancer cells through upregulating *PD-L1* gene expression. Consistently, we observed an inverse correlation between *PD-L1* and *TET2* expression levels regardless of whether the breast cancer cell lines were cultured in vitro (CCLE data) or in cancer samples from patients (TCGA data). Therefore, our work expands our current understanding of the pleiotropic role of TET2 loss in breast cancer pathogenesis, especially as it relates to immune evasion.

Interestingly, by analyzing multiple cancer cell lines from CCLE, we noticed that the correlation between TET2 and PD-L1 in lung cancer cell lines is also negative and more significant than that in breast cancer cell lines. This finding suggests that the TET2/PD-L1 negative regulatory axis may exist not only in breast cancer but also in other types of cancers. Given that anti-PD-1/L1 therapy has been successfully applied in the first-line therapy of lung cancer, it is of great interest to validate this observation in clinical patient samples and explore the prognostic value of TET2 expression in predicting the responsiveness of lung cancer patients to anti-PD-1/L1 therapy. If true, it may provide valuable clues and new strategies to improve immunotherapy treatment of lung cancer.

Multiple signaling pathways are aberrantly activated in the complicated tumor microenvironment of breast cancers. Among them, several signaling pathways (such as hypoxia, AMPK, IFN, and oxidative stress) have been reported to regulate TET2 expression level. The findings of our study suggest that the tumor microenvironment may modulate the expression of PD-L1 gene in breast cancer through targeting the TET2/HDAC complex. Given that anti-PD-1/L1 immunotherapy has been widely used in clinics, we speculate that the combination of HDAC inhibitors and targeting TET2 with anti-PD-L1 immunotherapy may be a new strategy for breast cancer patients who have low responsiveness to anti-PD-1/L1 immunotherapy. Currently, four HDAC inhibitors, Vorinostat, Romidepsin, Belinostat, and Panobinostat have been approved by FDA for cancer treatment [46]. Several studies have reported that HDAC inhibition make for increased PD-L1 expression in melanoma [47], ARID1A-inactivated Ovarian Cancer [48], and anaplastic thyroid cancer [49]. These findings, together with ours, reinforce a rationale for applying HDAC inhibitors or targeting TET2 to augment the immunotherapy of breast cancer.

## 5. Conclusions

In conclusion, our study provides clear evidences that *PD-L1* gene transcription is negatively regulated by the TET2-HDAC1/2 complex in breast cancer cells. Although more work remains to be done with regard to the regulatory mechanism and functional role of TET2/PD-L1 axis in the anti-tumor immunity, our findings suggest that targeting TET2 or HDAC1/2 might be a potential combination strategy for the anti-PD-1/PD-L1 immunotherapy treatment of breast cancer.

## Figures and Tables

**Figure 1 cancers-13-02207-f001:**
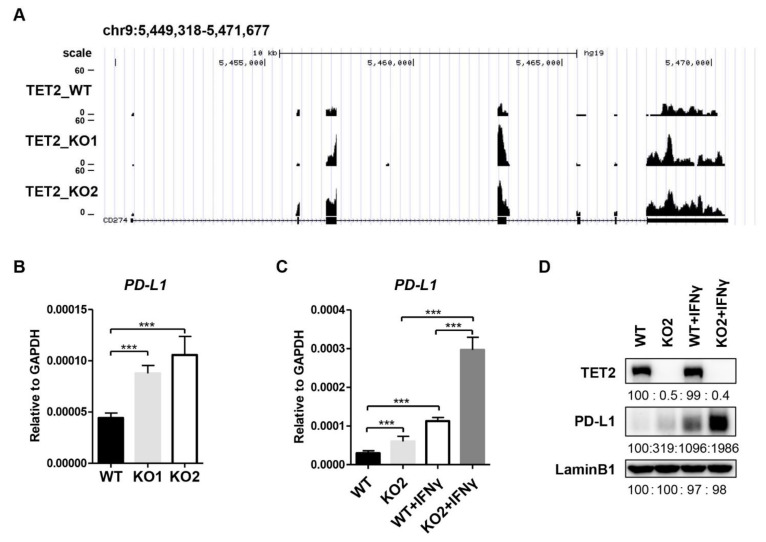
TET2 is a negative regulator of *PD-L1* gene transcription in breast cancer cells. (**A**) RNA-seq snapshot of *PD-L1* (*CD274*) gene expression in WT, *TET2*_KO1 and *TET2*_KO2 MCF7 cells. (**B**) RT-qPCR analysis of the relative mRNA expression levels of *PD-L1* in WT, KO1, and KO2 MCF7 cells. (**C**) RT-qPCR analysis of the relative mRNA expression levels of *PD-L1* in WT and *TET2* KO MCF7 cells with or without IFN-gamma (100 ng/mL) stimulation. (**D**) Western blot analysis of the PD-L1 protein levels in WT and *TET2* KO MCF7 cells with or without IFN-gamma (100 ng/mL) stimulation. (***, *p* < 0.001.)

**Figure 2 cancers-13-02207-f002:**
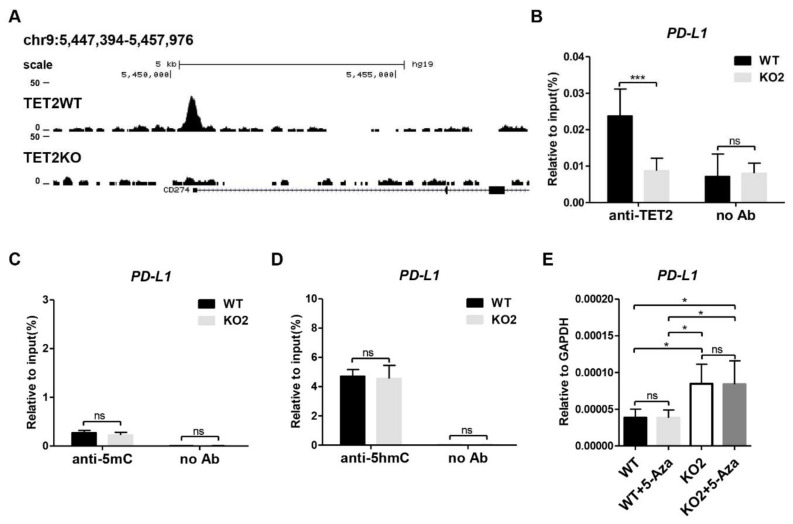
*TET2* KO does not alter the DNA methylation and hydroxymethylation level at the promoter region of *PD-L1* gene. (**A**) Snapshot of TET2 ChIP-seq data (GSE120756) at *PD-L1* (*CD274*) promoter in WT and *TET2* KO MCF7 cells. (**B**) ChIP-qPCR validation of TET2 occupancy at *PD-L1* promoter in WT and *TET2* KO MCF7 cells. (**C**) MeDIP-qPCR analysis of 5mC enrichment at *PD-L1* promoter in WT and *TET2* KO MCF7 cells. (**D**) hMeDIP-qPCR analysis of 5hmC enrichment at *PD-L1* promoter in WT and *TET2* KO MCF7 cells. (**E**) RT-qPCR analysis of the relative mRNA expression levels of *PD-L1* in WT and *TET2* KO MCF7 cells with or without 10 μM 5-Aza treatment. (*, *p* < 0.05; ***, *p* < 0.001; ns, not significant.)

**Figure 3 cancers-13-02207-f003:**
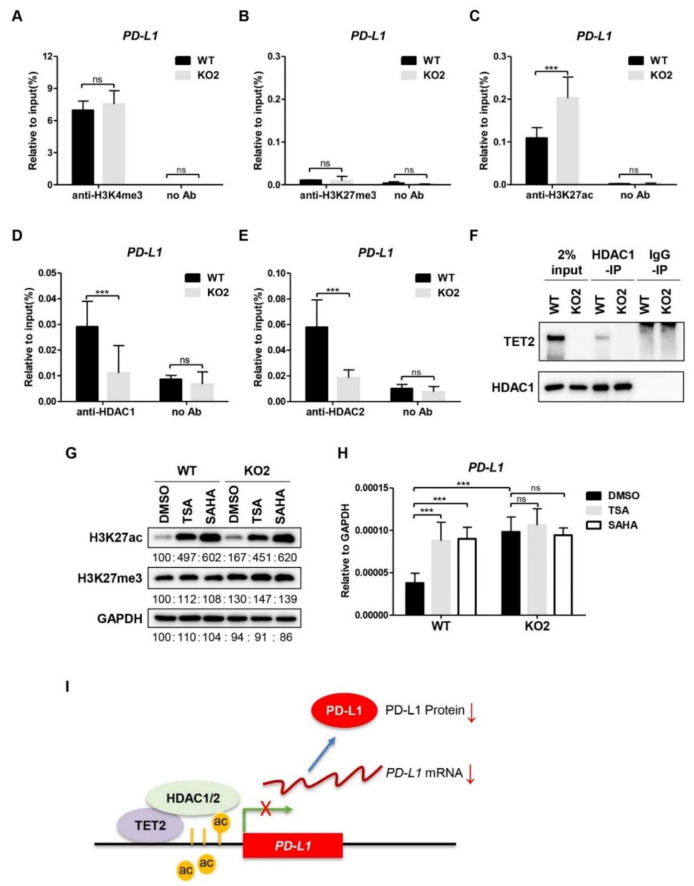
TET2 recruits HDAC1/2 to deacetylate H3K27ac at *PD-L1* promoter. (**A**) ChIP-qPCR analysis of H3K4me3 enrichment at *PD-L1* promoter in WT and *TET2* KO MCF7 cells. (**B**) ChIP-qPCR analysis of H3K27me3 enrichment at *PD-L1* promoter in WT and *TET2* KO MCF7 cells. (**C**) ChIP-qPCR analysis of H3K27ac enrichment at *PD-L1* promoter in WT and *TET2* KO MCF7 cells. (**D**,**E**) ChIP-qPCR analysis of the occupancy of HDAC1 and HDAC2 at the *PD-L1* promoter in WT and *TET2* KO MCF7 cells. (**F**) Western blot analysis of the anti-HDAC1-IP and IgG-IP products in WT and *TET2* KO MCF7 cells. (**G**) Western blot analysis of the global H3K27ac levels in WT and *TET2* KO MCF7 cells treated with or without HDAC inhibitors (TSA 1 μM; SAHA 5 μM). (**H**) RT-qPCR analysis of the relative mRNA expression levels of *PD-L1* in WT and *TET2* KO MCF7 cells treated with or without HDAC inhibitors (TSA 1 μM; SAHA 5 μM). (**I**) Schematic diagram of the working model in which TET2 inhibits *PD-L1* gene transcription through HDAC1/2-mediated histone deacetylation. (***, *p* < 0.001; ns, not significant.)

**Figure 4 cancers-13-02207-f004:**
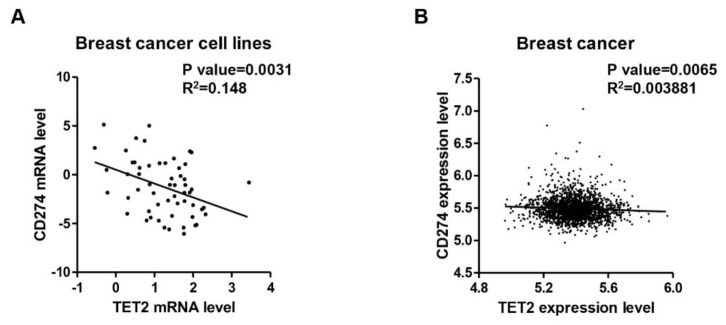
A negative correlation between *TET2* and *PD-L1* gene transcription in breast cancer. (**A**) CCLE data analysis showing a significant negative correlation between *TET2* and *PD-L1* expression levels in breast cancer cell lines (*n* = 57). (**B**) TCGA data analysis showing a mild but significant negative correlation between *TET2* and *PD-L1* expression levels in breast cancer tissues (*n* = 1904).

## Data Availability

The RNA-seq data presented in this study are openly available under the accession number GSE164032 (https://www.ncbi.nlm.nih.gov/geo/query/acc.cgi?acc= GSE164032), Submitted on 30 December 2020, Released on 1 June 2021.

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
