# Peer review of "TET2 Inhibits PD-L1 Gene Expression in Breast Cancer Cells through Histone Deacetylation"

_cancers, 2021, doi:10.3390/cancers13092207_

Round 1

Reviewer 1 Report

The manuscript has been fully revised.

Author Response

Thank you for your professional suggestions and help!

Reviewer 2 Report

The authors have reasonably dealt with my suggestions

Author Response

(The authors gave the same response as above.)

Reviewer 3 Report

Overall, I confirm the positive judgment on the quality of the manuscript and on the scientific soundness of the study described in it. However, I regret that the only objection I raised during the first revision was accepted by the authors as a request for explanations to the Reviewer, without further changes to the text. Given the multiple expertise of potential Cancers readers, it was implied that the required clarification should be included in the text.

Author Response

Thank you for your appreciation of our work and valuable comments! As you requested, we have clarified the rationale for using the DNMT inhibitor 5-Aza-CdR in 3.2 paragraph (page 6, line 221-227).

Round 2

Reviewer 3 Report

I recognize that the authors have made an effort to make the article more relevant according the reviewers’ comments and suggestions.

This manuscript is a resubmission of an earlier submission. The following is a list of the peer review reports and author responses from that submission.

Round 1

Reviewer 1 Report

The authors report that transcriptional repressor TET2 regulates the expression of PD-L1 in breast cancer cells. They showed that tumor suppressor TET2 recruits HDAC1/2 around the promoter region pf PD-L1 and facilitates the deacetylation of H3K27ac, followed by the control of PD-L1 expression. In conclusion, they proposed a new epigenetic regulation of PD-L1 in breast cancer. As authors referred several manuscripts, they concluded TET2 controls PD-L1 expression.

The overall data seems relevant, but authors should revise and provide the reliable data. In Figure 1E, authors used MDA-MB-231 cells for overexpression of TET2. Authors should mention why they used MDA-MB-231 cells, and possibly show overexpression of TET2 in KO2 MCF7.

Authors should mention “CD274” if they use it instead of PD-L1.

Is #A8182 true? #A8183? (Page 8, line 244)

Authors should correct the style of some manuscripts and references.

On Table S1, CCLE data analysis showed that correlation is negative on breast as well as lung cancers. Authors should mention regarding TET2 and PD-L1 in lung cancer in discussion for improvement of the manuscript.

Reviewer 2 Report

In this report, the authors have characterized the interplay between the expression of TET2 and PD-L1 as modulator of the immune response in the context of breast cancer. This study was mostly performed using the MCF7 cell line and an MCF7-TET2-KO, and some, not all, additional experiment with the MDA-MB-231. Finally, they have tested their observations in a panel of primary tumors.

In Figure 1, they show the effect of TET-KO on the levels of PD-L1. This figure should include Northern blots for the levels of mRNA from PD-L1 in both cells lines. The can be included in the same gel. This affects parts B to D. based on data in this Figure, elimination of TET2 is not enough, what it does is to prevent the response to induced by IFNy. The differences shown by qt-PCR range in two to three fold, while in the western blot D (not quantified is much larger).

Figure 1E, F. Experiments in MDA-MB231. The same discrepancy occurs between protein level and qt-RT-PCR. A PD-L1 mRNA reduction in half by qRT-PCR (F) results in a complete loss of protein (part E). A northern blot to detect the PD-L1 mRNA is necessary to clarify the discrepancy. Can an alteration of PD-L1 mRNA stability explain the discrepancy?

The western blots in this figure should be quantified.

Figure 2B. Inclusion of a western blot with anti-TET2 is necessary. The difference is only two fold, but what is the variation due to the amount of antibody in each of the immunoprecipitates?.

In Figure 2. The significant effects of TET2-KO on 5mC should also be shown by immunofluorescence. Is it a general effect or is it specific for the PD-L1 promoter?  If by IF there is no change in 5mC, it will indicate its effect is specific or restricted to few genes. If there is a general drop in 5mC levels, PD-L1 is just one more gene that is affected.

An experiment showing the methylated residues in the PD-L1 promoter will be very helpful. No differences in total methylation mean that the number methylated cytosine are the same in the two experimental conditions, wt and KO. It would be very interesting to know if they are located in similar or different C residues within the same area of the promoter.

Figure 3C. The western blot should include detection of H3K27me as control for comparison with DMSO.

Fig. 3H. They postulate that TET2 interacts with HDAC1/2. Have they really tested this hypothesis? To demonstrate that that is the case the authors should determine this interaction should be simple by performing a reciprocal immunoprecipitation experiment, done under different conditions (with and without IFNy treatment) to validate their hypothesis.

Fig. 4. The validation of the TET2-PD-L1 was performed in a group of breast cancer data. The prediction based on the MCF7 cells lines is borderline in the tumor panel. This might be a consequence of not analyzing the data based on three main breast cancer groups: ER/PR +, ERB2+ and triple negative. It is possible that a better correlation can be found if they reanalyze their data based on the three main breast cancer subtypes. Based on TableS2 there is a wide variation in the correlation between TET2 and PD-L1 expression among different types of breast cancers. Therefore, the analysis should be separated for each tumor subtype, better if the associate it to their specific genetic markers (ER, PR, HER2, triple negative). How do they explain the difference between luminal A and B. In Luminal A and basal-like the differences are not significant, but as breast cancer types they are very different. What is the explanation for that.

Most of the experiments have been done in MCF7 cells. The information on MDA-MB-231 should be included as supplementary. In that way, the results will be more convincing. Because they have mostly relied on MCF7, perhaps the clear correlation is less clear in the tumor panel.

The manuscript will improve significantly if they include a panel breast cancer cases and determine the levels of TET2 and PD-L1 by IHC or IF.

The regulatory role of TET2 on PD-L1 promoter is clear, but its real significance in the context of breast cancer cases is not clear.

Other comments:

Western blots do not show the position of size markers.

Uncropped gels are not included as supplementary information.

Reviewer 3 Report

Cancer immunotherapy has emerged in the past years as a promising therapeutic options also for solid tumors, including breast cancer. Since the well known role of the PD-1/PD-L1 axis in tumor immune escape, many immunotherapeutic strategies have targeted such immune checkpoint molecules. The present study is aimed at investigate the transcription regulatory mechanisms of PD-L1 gene, with particular reference to the DNA demethylase TET2. Surprisingly, the Authors found that TET2 acts as PD-L1 gene suppressor in breast pre-clinical cellular models, through the recruitment of histone deacetylases HDAC1 and HDAC2. Such an evidence, opens novel therapeutic options for breast cancer, involving a potential combined therapy between approved anti-PDL1 antibodies and HDAC inhibitors.

The manuscript appears to be an extensive, well-written and well-conceived study on the matter, endowed with clarity, accuracy, consistency and balance. Particular emphasis has been devoted to the mechanisms of epigenetic regulation of PD-L1 gene expression.

  In addition to molecular details about the rationale of such potential therapeutic approaches in breast cancer cell lines, the Authors have included in silico analysis showing a negative correlation between TET2 and PD-L1 in breast cancer tissues from patients (TCGA data). I only suggest the Authors to better describe the rationale for using the DNMT inhibitor 5-Aza-CdR in 2.2 paragraph (line 116, page 4, Figure E).
